# Evaluation of various methods of selection of *B. subtilis* strains capable of secreting surface-active compounds

**Beata Koim-Puchowska, Grzegorz Kłosowski** \*, **Dawid Mikulski, Aleksandra Menka**

Kazimierz Wielki University, Department of Biotechnology, Bydgoszcz, Poland

\* klosowski@ukw.edu.pl

## Abstract

The aim of the study was the evaluation of a three-step method for the selection of bacterial strains capable of producing surfactin. The procedure consisted of the following steps: 1. blood agar test, 2. measurement of the surface tension (ST) of the medium using the du Nouy method before and after submerged culture, 3. qualitative and quantitative assessment of surfactin by HPLC. Forty five *Bacillus subtilis natto* strains producing haemolysis zones ($\geq$3mm) were selected. Nineten of them reduced ST of the medium to $\leq$ 40 mN/m; in six cases, the reduction was as much as 50%. All indicated strains produced surfactin. Positive correlations (p <0.5) between the percentage reduction of ST of the medium and surfactin concentration (r = 0.44), indicate that this parameter is determinant of the ability to synthesize this compound. The blood agar test has been shown to be useful only as a pre-selection criterion for surfactin producers (18 strains selected by this method reduced ST by only $\leq$30%). The proposed selection strategy proved effective and made it possible to select the BS15 strain that reduced the ST of the medium to 30.56 ± 0.15 mN/m and simultaneously provided a high concentration of surfactin compared to other strains.

## Introduction

Surfactants (surface active agents) are a group of compounds widely used in industry, agriculture and in households as a component of cleaning agents, medicines and cosmetics [1]. The global market for surfactants has been estimated at approximately USD 30.65 billion in 2015. It is forecasted that due to the continuous upward trend (around 4.4% per year) it may reach USD 39.69 billion by 2021 [2]. The majority of synthetic surfactants are produced using chemical methods from petrochemical raw materials, which poses a toxicological threat to living organisms, especially in aquatic ecosystems [3–5]. An alternative production method is biosynthesis carried out with the use of specialized microorganisms capable of producing biosurfactants with lower toxicity, high resistance to extreme pH, temperature or salinity, and, most importantly, increased biodegradability [6–11]. Due to their ability to reduce surface tension (at the water-air interface) and interfacial tension (at the water-oil interface), biosurfactants are among the most versatile process chemicals [12,13]. Their surface-active properties result

**Funding:** This study was funded by the Polish Minister of Science and Higher Education, under

the program "Regional Initiative of Excellence" in 2019–2022 (Grant No. 008/RID/2018/19). The funders had no role in study design, data collection and analysis, decision to publish, or preparation of the manuscript.

from the amphiphilic structure of the molecule, which favors the aggregation of these compounds at the interface [14,15]. The decrease in surface tension is caused by the increase in biosurfactant concentration and the formation of aggregated amphipathic molecules, so-called micelles. The surfactant concentration at which the micelles are formed is referred to as the critical micellization concentration (CMC) [11,16]. This parameter corresponds to the minimum surfactant concentration that is necessary to achieve the maximum reduction of surface tension [11,16,17]. The CMC value indicates the efficiency of the surface active compound, while its effectiveness is related to the measurement of surface tension and interfacial tension [16]. Surfactin is one of the most effective surfactants with a high application potential [18–21]. This compound is an interesting bioproduct due to its properties. Surfactin shows antibacterial activity towards *Mycoplasma sp.* [22], it is an antineoplastic agent for breast, colon cancer and leukemia [23]. It also exhibits emulsifying effect and thus accelerates biodegradation of hydrocarbons in oil contaminated areas [24]. Therefore, surfactin, a lipopeptide, can successfully replace synthetic compounds in many branches of the economy. This biosurfactant reduces the surface water tension from 72 to 27 mN/m and the interfacial tension of water/n-hexane from 43 mN/m to 1 mN/m [10,25,26]. In addition, the CMC of surfactin is very low, i.e. 0.017 g/L in water [27]. The surfactin properties listed above result from its chemical structure [28]. This compound consists of a peptide loop of seven amino acids (L-Glu, L-Leu, D-Leu, L-Val, L-Asp, D-Leu, L-Leu), and a hydrophobic fatty acid chain thirteen to fifteen carbons long. L-Glu, D-Leu, L-Asp, D-Leu amino acids are the molecular backbone of the compound while others can be exchanged for L-Val L-Ile, L-Leu or L-Ala [29]. Hence, surfactin is produced by microorganisms as a mixture of closely related isoforms that differ in the length and branching of the fatty acid side chains as well as in the configuration of amino acids in the peptide ring. The structure of surfactin, which depends on the composition of the culture medium and the specifics of the producer himself, largely determines the surface-active properties of this compound [20,30,31]. Surfactin, like other biosurfactants, is produced as a secondary metabolite in the stationary phase of microbial growth [28,32]. It is not necessary for the growth and development of microorganisms, but its synthesis may increase their adaptability to given environmental conditions and provide an alternative defense mechanism [33].

A special role in the biosynthesis of lipopeptide biosurfactants is played by bacteria of the genus *Bacillus (B. subtilis, B. pumilus, B. myvensis, B. licheniformis, B. amyloliquefaciens).* These bacteria produce peptide synthetases and/or polyketide synthetases and synthesise surfactin derivatives with different chemical structure and properties [20,34,35]. It should be noted that most reports on surfactin biosynthesis refer to *Bacillus subtilis* as its main producer [20,36,37]. Despite numerous studies on surfactin biosynthesis since 1968, the production of this compound on an industrial scale still involves large financial expenses. The costs of media components can account for up to 50% of production expenditure while the efficiency of microbiological biosynthesis is still low [38–40]. This is a significant obstacle limiting the spread of this technology and commercial use of biosurfactants [10]. In order to develop a highly efficient surfactin biosynthesis process using *Bacillus sp.*, effective selection strategies should first be developed to obtain strains capable of biosynthesis of isoforms that significantly reduce surface tension. So far several producers have been identified, mainly within the *Bacillus subtilis* species, including ATCC 21332, DSM 3256, DSM 3258 strains. The surfactin yield (g/L) reported for these strains cultivated on synthetic media was ca. 2.39 ± 0.9; 1.79 ± 0.8, 1.6 ± 0.11, respectively [29]. However, the yield of surfactin produced by a strain depends to a large extent on the culture conditions and methods, and the availability of nutrients in the medium [12]. It should be emphasized that the strains deposited in collections account for only 1% of the total microflora living in the natural environment [41]. The search for new native strains, acquiring and developing knowledge about their physiology, metabolism and

genetics is the key to an efficient microbial bioconversion of food substrates (especially by-products of agro-food processing) to surfactin isoforms [12,28,34,36,42,43]. Effective screening of microorganisms requires well-chosen experimental and analytical methods that guarantee a quick and efficient assessment of metabolic characteristics of isolated strains. So far, many methods have been developed for the qualitative and quantitative selection of microorganisms in the production of biosurfactants, including the measurement of surface, interfacial and emulsifying activity of bioproducts, oil spreading method, drop collapse test or more advanced, verifying the products, such as high performance liquid chromatography [44,45]. It should be noted, however, that only their proper combination at subsequent stages of the selection enables efficient and low-budget estimation of the ability of surfactin biosynthesis by native strains.

The goal of this study was to evaluate the effectiveness of selected methods for screening *Bacillus subtilis* strains isolated from fermented food products for the production of surfactin isoforms. The ability of surfactin biosynthesis was verified during three-stage selection. First, the isolated strains were evaluated for the intensity of erythrocyte lysis on solid media containing defibrinated sheep blood (blood agar test), which indicated the likely biosurfactant biosynthesis capabilities. Next, strains forming hemolysis zones were used for the biosynthesis of surfactants on model liquid medium (SmF), and the surfactin biosynthesis ability was evaluated on the basis of the degree of reduction of the surface tension (ST) of the medium after cultivation. The final step in strain selection, which verified whether screening was successful, was confirmation of the ability to synthesize surfactin isoforms using high performance liquid chromatography. During the study, an attempt was made to assess the relationship between the concentration of surfactin produced by isolated strains and the percentage of surface tension reduction in the post-culture medium.

## Materials and methods

### Isolation of *Bacillus subtilis natto* strains. Blood agar test

The bacteria were isolated from a natto food product (Ton Color, Poland), whose basic ingredient is soya beans subjected to prior fermentation by *Bacillus subtilis natto* strains. The material (approximately 5 g) was homogenized in a Stomacher bag, equipped with a side membrane, with the addition of 20 ml of sterile 0.9% NaCl. One ml of the filtrate was taken and a series of 10-fold dilutions were performed in sterile 0.9% NaCl in the range of $10^{-1}$ to $10^{-7}$. Samples of $10^{-6}$ and $10^{-7}$ dilutions were plated on agar medium supplemented with 5% defibrinated sheep blood (Columbia Blood Agar, Poland). The medium was prepared by dissolving 43 g of dry base mixture (Columbia Agar Base, Grasso Biotech, Poland) containing casein hydrolyzate (5.0g), meat extract (8.0g), yeast extract (10.0g), sodium chloride (5.0g), corn starch (1.0g) and agar (14.0g) in 1L distilled water, while stirring and heating the solution to 90˚C. Once the components have completely dissolved, the medium was sterilized in an autoclave for 15 min at 121˚C, and then enriched with 50 ml of defibrinated sheep blood under aseptic conditions. The pH of the medium was 7.3 ± 0.2. The plates after inoculation were incubated for 96 h at 30˚C. The strains showing the ability to erythrocyte lysis, manifested by the occurrence of clear zones (> 3mm) around the colonies, had been deposited on agar slopes (bacteriological tryptone peptone 5g/L, 2.5 g/L yeast extract, 1g/L glucose, 15g/L agar, pH 7.2–7.4) and subsequently incubated for 96 h at 30˚C. All isolated strains were deposited in cryobanks (Grasso Biotech, Poland) and stored at -20˚C until analysis. Differences between isolated *B. subtilis* strains were confirmed by analysis of allele sequence variability in the pta (phosphate acetyltransferase) locus using the MLST (multilocus sequence typing) method.

## Submerged fermentation (SmF) conditions

The SmF of the isolated strains was carried out in two stages. In the first step, an inoculum was obtained, which was transferred to the mineral medium [46] enabling the biosynthesis of surfactin. In order to obtain an inoculum, 100 ml of nutrient broth (bacteriological tryptone peptone 5 g/L, yeast extract 2.5 g/L, glucose 1 g/L, pH 7.2–7.4) was inoculated under aseptic conditions with selected bacterial strains. Then the culture was grown for 24 h with shaking (70 rpm) in a water bath at 37°C. Next, 2.5 mg of inoculum obtained in this way was transferred to Cooper's medium (glucose 40g/L; $NH_4NO_3$ 4g/L; $KH_2PO_4$ 4.08 g/L; $Na_2HPO_4$ x2$H_2O$ 7.12g/L; $MgSO_4$ x7$H_2O$ 0.2 g/L; $CaCl_2$ 0.0008g/L; $FeSO_4$ x 7$H_2O$ 0.0011g/L; EDTA 0.0012 g/L, pH 7.0). SmF cultures with shaking (70 rpm) were carried out in triplicate in 250 ml flasks for 120 h at 37°C [46].

## Measurement of biomass concentration

In order to determine the biomass concentration of selected *Bacillus subtilis natto* strains after the culture, 5 ml of cell suspension was taken and centrifuged twice (10 min, 8000 g, MPW-260R centrifuge, MPW-Med Instruments, Poland); the precipitate was rinsed with 5 ml of 0.9% NaCl. Finally, cell biomass was suspended in 5 ml of 0.9% NaCl and the optical density ($OD_{600}$) was measured using an UV-VIS spectrophotometer (Pharo 300, Merck) [40]. The biomass concentration of the analyzed strains [mg/ml] was determined using a curve showing the dependence of the optical density (OD) of liquid test culture (*Bacillus subtilis* no. ŁO820; from the Collection of pure cultures of industrial strains, Łódź University of Technology, Poland) on the amount of dry cell mass. The biomass was dried to a constant mass using a weighting dryer (RADWAG, WPS-30S) at 105°C and 20 s sampling time.

## Measurement of surface tension

The surface tension of the culture medium before and after cultivation was measured by the Du-Nouy-Ring method [44] using a tensiometer (model PI-MT1M, Donserv, Poland). This method measured the force required to detach the platinum ring of radius 20.5 mm from the surface of the culture medium at room temperature. Between the measurements, the platinum ring was rinsed with ethanol and then allowed to dry. All measurements were taken in the post-culture medium after biomass removal by centrifugation (2400 g, 15 min, 4°C). In order to increase the representativeness of the results, the measurements were repeated five times.

## Extraction and determination of surfactin concentration

Qualitative and quantitative assessments of surfactin isoforms were preceded by extraction using affinity chromatography with the solid-phase extraction (SPE) system. For extraction we used Bond Elut C18 columns (Agilent Technologies), which are retentive for non-polar compounds. Surfactin was extracted from the post-culture medium after removing the biomass of bacteria (2400g, 15 min, 20°C). Prior to the extraction, the SPE column was conditioned with methanol and stabilized with distilled water (as recommended by the manufacturer). Surfactin was extracted by applying 20 mL of culture medium to the SPE column. Isolation was carried out at a flow rate of 1 drop per second. The column bed was then rinsed with 5% methanol and dried thoroughly under reduced pressure (SPE system, Agilent Technologies). Elution of surfactin was carried out using HPLC grade methanol. Before injection, the solution was filtered through a membrane filter (pore size 0.22 μm). The chromatographic separation was performed by HPLC (Model 1260 chromatograph, Agilent Technologies) with DAD detection, according to the method presented by [29]. The Poroshell 120 EC-C18 column (4.5 x 150 mm,

2.7 μm) equipped with a guard column (Poroshell 120 EC-C18, 3.0 x 5 mm, 2.7 μm) was used. Acetonitrile and 3.8 mM trifluoroacetic acid (80:20, v/v) were used as the mobile phase with isocratic flow of 1 mL/min at 25˚C. Surfactin isoforms were detected at 205 nm (peptide bond absorption wavelength). Quantitative calculations were performed using the external standard method (ESTD) with methanolic surfactin solution obtained from *Bacillus subtilis* (Sigma-Aldrich).

### Statistical analysis

Experiments were performed in triplicate (n = 3). The results are presented as mean ± standard deviation (SD). Data analysis was carried out using the Statistica software ver. 13.3.

### Results and discussion

The interest in lipopeptide surfactants, due to their wide application potential, forced the development of many screening methods that could verify potential producer strains. Because these methods have various constraints, e.g. low selectivity or specificity for a selected group of biosurfactants, only their proper combination in a specific strategy can be successful, i.e. allow for acquisition of completely new strains capable of producing biosurfactants [44,45]. The strategy for selection of *Bacillus subtilis natto* strains synthesizing surfactin proposed in this paper consisted of three key stages whose main objective was a fast and effective screening. In the first stage, we investigated the ability of isolated strains to produce hemolysis zones on a solid medium enriched with defibrinated sheep blood after 96 h incubation at 30˚C, because the lysis of erythrocytes by the strain being analyzed is considered to give an indication on biosurfactant production [47]. At this stage, 45 strains having clear zones (> 3 mm) around the colonies were selected (Fig 1).

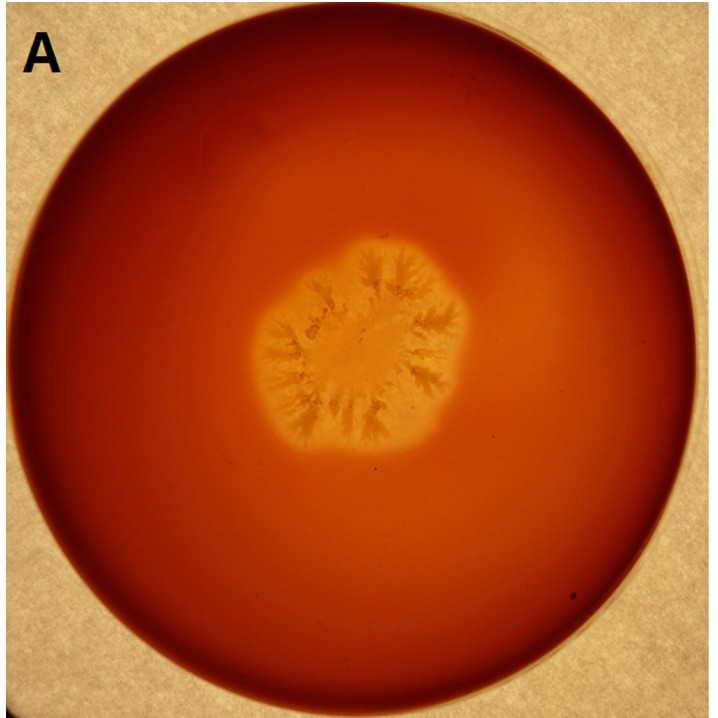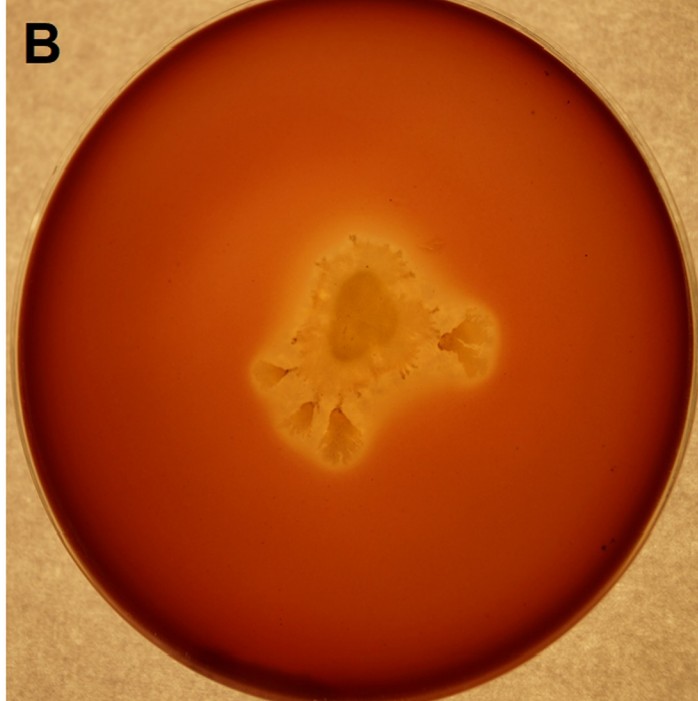

**Fig 1.** Hemolysis zone around a single BS22 (A) and BS25 (B) colony on a solid medium with 5% defibrinated sheep blood (Columbia Blood Agar).

The next step in verifying selected strains as potential surfactin producers was the measurement of surface tension (ST) of Cooper's medium, before and after the submerged culture (SmF). This method was widely used in many studies as a measure of the surface-active properties of microbiologically synthesized biosurfactants [43,48,49]. Surface tension measurements performed in our study clearly indicated a decrease in the value of this parameter by about 50%, i.e. to ca. 30 mN/m, in post-culture media of as many as 6 strains: BS15, BS19, BS 21, BS25, BS35, BS36 (Table 1,Fig 2).

Earlier reports [44,50] suggested that strains that have the ability to reduce the medium surface tension to ≤ 40 mN/m are considered potential biosurfactant producers. Based on this assumption, it should be concluded that approximately 38% of strains selected by the authors of the present study met this criterion. The surface tension of the post-culture medium of the six selected strains was close to 30 mN/m. De Faria et al. [51], Bezza and Chirwa [52], Al Wahaibi et al. [53], Jha et al. [54] also confirmed the ability of the studied microorganisms to lower the ST of the culture medium to this level. On the other hand, about 18 strains of the 45 strains selected in our study showed a decrease in surface tension of the culture medium by only ≤ 30%. This indicates that the ability to form hemolysis zones can not be the only determinant taken into account when assessing the capability to produce biosurfactants by *Bacillus subtilis natto* strains. Youssef et al. [55] demonstrated that as much as 60% of 205 strains showing erythrolytic capacity reduced ST only to > 60 mN/m. Meanwhile, despite the negative blood agar test, 38% of strains reduced the ST even to ca. 35 mN/m. It was also believed that low correlation (r = -0.15) between surface tension and the occurrence of haemolysis zones proved poor efficiency of this method in the verification of strains capable of producing surfactin. Joshi et al. [43], Hsieh et al. [47], Mulligan et al. [56] concluded that blood agar test can only be used as a preliminary screening of strains, enabling identification of potential surfactant producers. This was due to the limitations of this method resulting from its low specificity associated, eg, with the diffusion of biosurfactants in the agar medium or the action of lytic enzymes (including proteases), which might suggest false negative or false positive results [44]. However, its advantages, i.e. speed (96 h) and low costs, speak for its use in the initial stage of selection.

The final stage of the screening was verification of earlier stages of the selection, i.e. determination of the concentration of surfactin and its isoforms in the medium. HPLC is considered one of the best methods for qualitative and quantitative characterization of surfactin, and therefore it has been used in many studies on microbiological surfactin biosynthesis [29,32,36,40,57]. In our study, 19 strains were selected for the assessment. The selection criterion was the ability to reduce the surface tension by ≥ 35%. The strains BS14 and BS20, which reduced the ST of the medium to a lesser extent (by ca. 32%) were also taken into account. They were a reference point for confirming the hypothesis that there is a relationship between the concentration of surfactin and the percentage decrease in surface tension of the culture medium. Due to the need to compare the analyzed strains for the ability to surfactin biosynthesis, the final results, i.e. the concentration of this compound, are expressed in μg/mg of biomass. Although the average biomass concentration of the tested strains after cultivation was 0.75 ± 0.073 mg/ml, the differences in this parameter between the strains were considerable. For example, while the biomass concentration in the post-culture medium of the BS1 strain was 0.37 ± 0.081 mg/ml, for the BS40 strain it was 1.37 ± 0.047 mg/ml (Table 1). The rate of biomass production is related to the differences in the assimilation of nutrients and results from the adaptability of particular strains to the given culture conditions.

We observed considerable differences between the assessed strains in the surfactin producing abilities. The concentration of the compound produced ranged from 0.82 ± 0.40 μg/mg (BS45) to 30.40 ± 1.44 μg/mg (BS15) (Fig 3).

**Table 1. Average biomass concentration of selected strains after submerged culture (SmF).** Average percentage decrease in the surface tension of the culture medium during the cultivation.

| Strains | Biomass concentration [mg/ml] | | | Surface tension decrease [%] | | |
| --- | --- | --- | --- | --- | --- | --- |
| | Mean | ± | SD | Mean | ± | SD |
| BS1 | 0.37 | ± | 0.081 | 13.74 | ± | 7.130 |
| BS2 | 1.20 | ± | 0.053 | 21.23 | ± | 0.335 |
| BS3 | 0.63 | ± | 0.189 | 40.54 | ± | 10.784 |
| BS4 | 0.80 | ± | 0.050 | 22.37 | ± | 0.598 |
| BS5 | 0.84 | ± | 0.171 | 15.73 | ± | 0.179 |
| BS6 | 0.84 | ± | 0.177 | 18.56 | ± | 2.994 |
| BS7 | 0.95 | ± | 0.246 | 36.84 | ± | 7.956 |
| BS8 | 0.80 | ± | 0.080 | 4.53 | ± | 1.102 |
| BS9 | 0.44 | ± | 0.013 | 30.82 | ± | 1.256 |
| BS10 | 0.44 | ± | 0.007 | 33.75 | ± | 0.478 |
| BS11 | 0.80 | ± | 0.007 | 32.26 | ± | 1.854 |
| BS12 | 1.02 | ± | 0.181 | 29.27 | ± | 2.346 |
| BS13 | 1.10 | ± | 0.017 | 14.82 | ± | 1.242 |
| BS14 | 0.58 | ± | 0.089 | 32.39 | ± | 2.843 |
| **BS15** | **0.46** | **±** | **0.008** | **52.77** | **±** | **0.404** |
| BS16 | 0.81 | ± | 0.031 | 28.01 | ± | 0.311 |
| BS17 | 0.51 | ± | 0.044 | 32.78 | ± | 1.048 |
| BS18 | 0.98 | ± | 0.027 | 24.93 | ± | 2.905 |
| **BS19** | **0.78** | **±** | **0.058** | **49.34** | **±** | **1.756** |
| BS20 | 0.75 | ± | 0.075 | 31.18 | ± | 13.444 |
| **BS21** | **0.76** | **±** | **0.002** | **54.12** | **±** | **0.229** |
| BS22 | 1.01 | ± | 0.084 | 35.81 | ± | 1.373 |
| BS23 | 0.57 | ± | 0.011 | 41.99 | ± | 6.636 |
| BS24 | 0.60 | ± | 0.062 | 42.75 | ± | 1.931 |
| **BS25** | **1.00** | **±** | **0.381** | **53.69** | **±** | **0.750** |
| BS26 | 0.55 | ± | 0.045 | 28.34 | ± | 1.733 |
| BS27 | 0.41 | ± | 0.007 | 28.34 | ± | 3.837 |
| BS28 | 0.44 | ± | 0.062 | 10.76 | ± | 3.121 |
| BS29 | 0.61 | ± | 0.012 | 13.66 | ± | 7.577 |
| BS30 | 0.26 | ± | 0.028 | 35.29 | ± | 5.478 |
| BS31 | 0.32 | ± | 0.004 | 3.69 | ± | 3.694 |
| BS32 | 0.72 | ± | 0.030 | 42.39 | ± | 2.042 |
| BS33 | 1.33 | ± | 0.086 | 20.89 | ± | 4.331 |
| BS34 | 1.01 | ± | 0.187 | 40.22 | ± | 6.064 |
| **BS35** | **0.67** | **±** | **0.001** | **49.44** | **±** | **1.420** |
| **BS36** | **0.75** | **±** | **0.079** | **49.44** | **±** | **2.037** |
| BS37 | 0.69 | ± | 0.021 | 46.23 | ± | 0.185 |
| BS38 | 0.71 | ± | 0.031 | 45.99 | ± | 2.284 |
| BS39 | 1.10 | ± | 0.277 | 46.25 | ± | 0.939 |
| BS40 | 1.34 | ± | 0.047 | 31.17 | ± | 4.535 |
| BS41 | 1.13 | ± | 0.035 | 29.05 | ± | 7.118 |
| BS42 | 0.64 | ± | 0.033 | 45.49 | ± | 1.314 |
| BS43 | 0.52 | ± | 0.062 | 43.49 | ± | 3.066 |
| BS44 | 0.52 | ± | 0.006 | 21.30 | ± | 4.305 |
| BS45 | 0.89 | ± | 0.109 | 44.37 | ± | 1.815 |

Data presented as mean ± SD for each strain (n = 3).

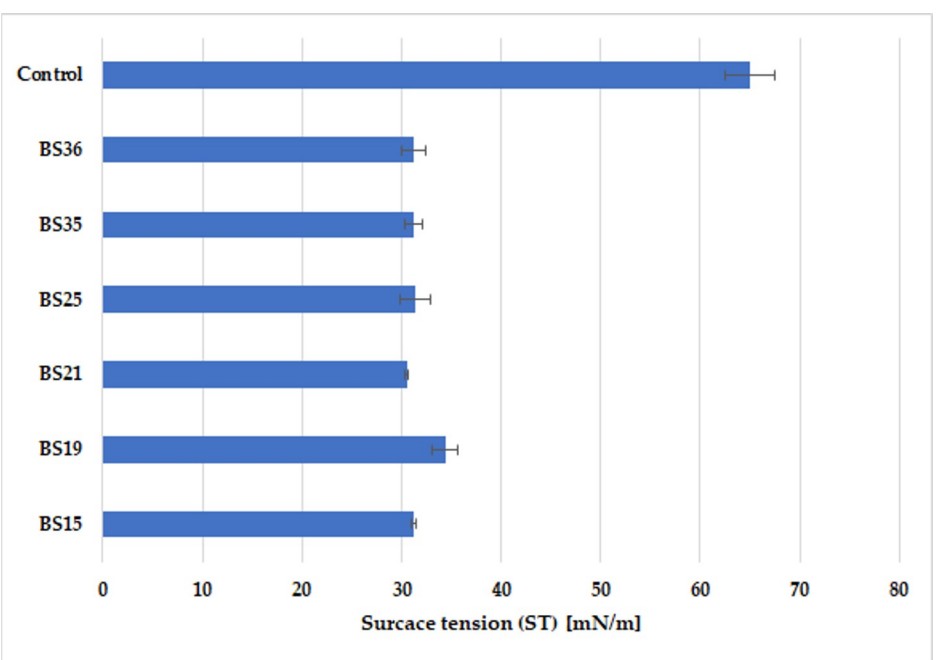

**Fig 2. Surface tension (ST) [mN/m] of Cooper's mineral medium measured before (Control) and after submerged cultivation (SmF) of selected *Bacillus subtilis natto* strains (BS15, BS19, BS21, BS25, BS35, BS36).**

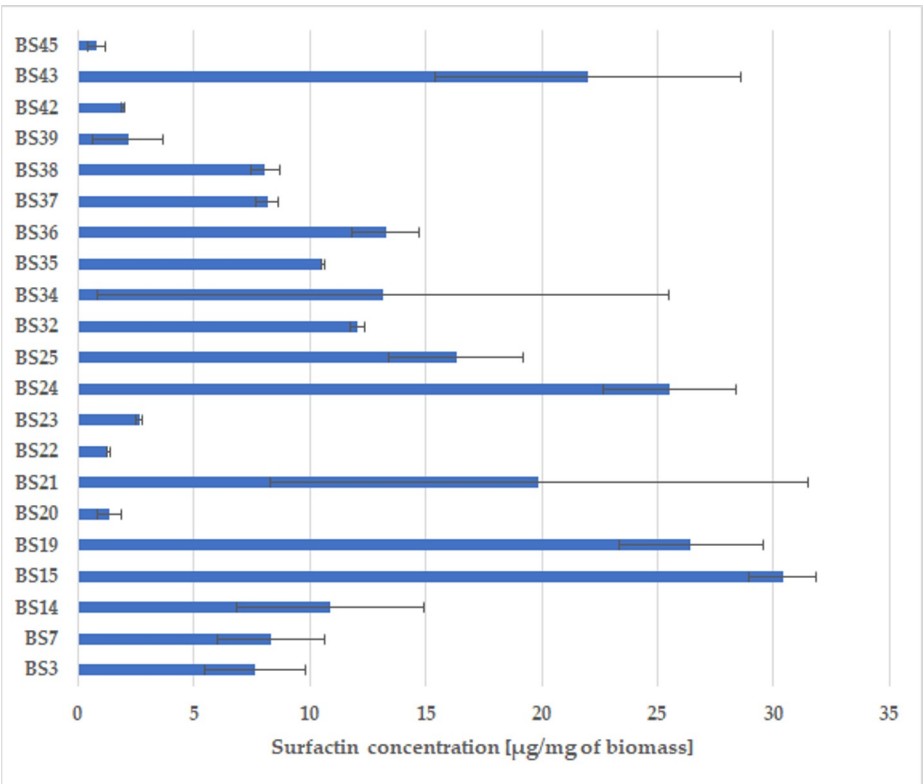

**Fig 3. Average concentration of surfactin [μg/mg] synthesized by selected strains (n = 3).** Error bars represent standard deviation (SD).

Particularly high concentration of synthesized surfactin (apart from the BS15 strain) was also found for the following strains: BS19 (26.45 ± 3.08 µg/mg); BS24 (25.54 ± 2.88 µg/mg) and BS43 (22.00 ± 6.60 µg/mg). However, the standard deviation (SD) of surfactin productivity for BS15 did not exceed 5%, which indicated a good repeatability of the results and distinguished this strain from the others. Thus, the application of the solid phase extraction (SPE) system along with the affinity chromatography and HPLC analysis of the eluate undoubtedly enabled rapid verification of the surfactin biosynthesis capacity. Thanks to the use of SPE columns for the extraction of hydrophobic compounds, the analysis time was significantly shortened and the risk of incorrect results that might occur during the multi-stage isolation procedures was minimized.

Because the main goal of this study was to evaluate the methods that enable the selection of potential surfactin producers, at this stage of the research we have abandoned the assessment of the maximum surfactin production efficiency of the obtained strains, which could be achieved by optimizing the culture parameters. Therefore, it should be emphasized that surfactin yields resulting from the production optimization reported by other authors, should not be directly compared with the surfactin production efficiency observed in this study. In order to maximize surfactin production, many authors Abdel-Mawgoud et al. [8], Amani et al. [58], Jokari et al. [59] adapted the culture conditions (pH, temperature, and especially dissolved oxygen concentration) and the composition of the medium specifically for a given *Bacillus subtilis* strain. Willenbacher et al. [21] showed that the modification of the mineral Cooper's medium composition by changing the glucose concentration (8 g/L), supplementing with sodium citrate (0.008 mM) and replacing the nitrogen source with $(NH_4)SO_4$ significantly improved the yield of surfactin produced by *Bacillus subtilis* DSM10$^T$ from 0.7 to 1.1g/L. In turn, Hsieh et al. [47] observed that the reference strain ATCC 21332, considered to be an outstanding surfactant producer (even 800 mg/L), was able to produce only 109.5 mg/L on the mineral medium if the fermentation products were constantly removed and the medium was enriched with metal cations. Jajor et al. [40], after testing two strains of *Bacillus subtilis*, #309 and KB1 *natto* for the production of surfactin under varying oxygen availability, demonstrated that increased oxygenation reduced surfactin biosynthesis in the culture of strain #309, while this factor acted as a stimulant on the strain isolated from natto.

The results indicate a significant (p < 0.05) positive correlation between the percentage reduction of surface tension and the total concentration of surfactin and its isoforms in the post-culture medium of selected strains (Table 2).

Thus, the percentage decrease in ST can be used as a reliable indicator of the ability of strains to produce surfactin, which would significantly accelerate the selection of potential

**Table 2. Relationship between the total concentration of surfactin and its isoforms A-F [µg/mg] and the surface tension decrease [%] of the medium after cultivation of *Bacillus subtilis natto* strains (n = 63).**

| Surfactin concentration [µg/mg] | Surface tension decrease [%] | |
|---|---|---|
| | r | p |
| Total | 0.44 | 0.000 |
| Isoform A | 0.42 | 0.001 |
| Isoform B | 0.51 | 0.000 |
| Isoform C | 0.37 | 0.003 |
| Isoform D | 0.42 | 0.001 |
| Isoform E | 0.43 | 0.001 |
| Isoform F | 0.46 | 0.000 |

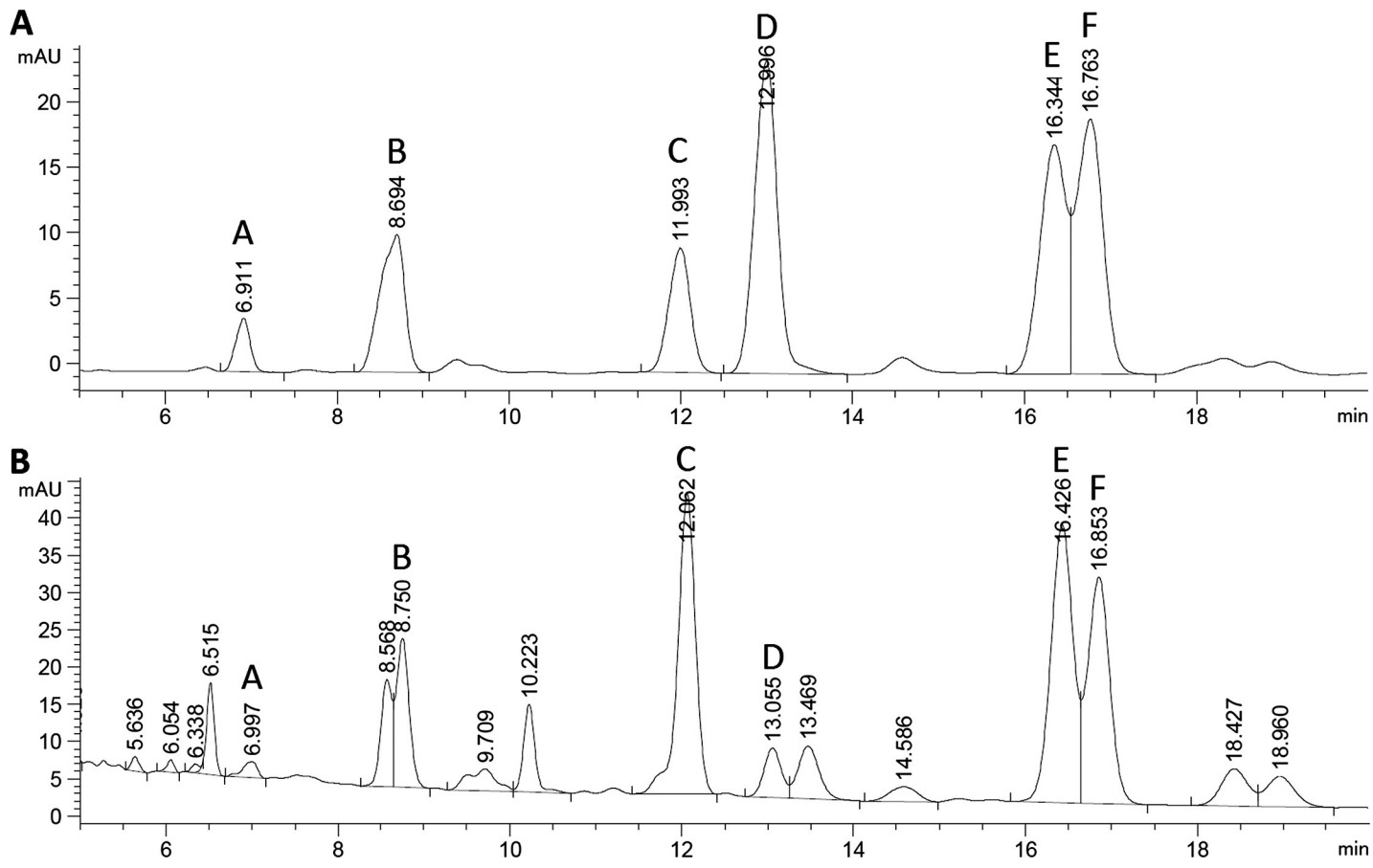

**Fig 4. Chromatogram of surfactin standard (from *Bacillus subtilis*, ≥95%, HPLC, Sigma-Aldrich, dissolved in methanol) against the chromatogram of surfactin synthesized by strain BS15.**

producers. There are, however, some limitations in the measurement range, as ST decreases with the increase of the biosurfactant concentration only until it reaches the critical micellization concentration (CMC). Once this value is reached, it is not possible to determine the increase in the concentration of surfactant. As a consequence, two strains may reduce ST to critical level and at the same time may differ significantly in terms of surfactin production efficiency [44], which is also evidenced by the results presented here. For example, BS15 and BS25 reduced the ST of the medium by about 53%, whereas the concentration of surfactin synthesized by them was 30.44 ± 1.44 μg/mg and 16.33 ± 2.9 μg/mg, respectively (Table 1, Fig 3).

Reports show that surfactin may exist in the form of five [40], six [60], eight [61] and even nine isoforms [62], differing in physicochemical properties. A chromatogram of the surfactin standard (≥95%) produced by *Bacillus subtilis* (Sigma-Aldrich) indicates the presence of 6 isoforms: A-F (Fig 4). Over 80% of selected strains having the ability to lower the ST of the medium by ≥ 35% (including BS15) synthesized all these isoforms (Table 3, Fig 5).

However, the percentages of the individual isoforms of the standard and of those produced by the analyzed *Bacillus subtilis natto* strains were not identical. The C isoform had the highest percentage (35%) of total concentration of surfactin produced by selected *B. subtilis natto* strains, followed by E (31%), F (19%), B (11%), D (3%),and A (1%) (Fig 5). The chromatogram of the standard showed that the D isoform had the largest percentage, followed by the isoforms F, E, B, C, and A. Interestingly, of the remaining 20% of the 19 selected strains, only BS20 and

**Table 3. Average concentration of surfactin A-F isoforms [μg/mg].** Data presented as mean ± SD for each strain (n = 3).

| Strains | Isoform A | Isoform B | Isoform C | Isoform D | Isoform E | Isoform F |
| --- | --- | --- | --- | --- | --- | --- |
| | Mean ± SD | Mean ± SD | Mean ± SD | Mean ± SD | Mean ± SD | Mean ± SD |
| BS3 | 0.14±0.03 | 0.66±0.26 | 2.34±0.56 | 0.16±0.14 | 2.67±0.52 | 1.67±0.73 |
| BS7 | 0.06±0.02 | 1.26±0.35 | 2.63±0.72 | 0.21±0.06 | 2.90±0.80 | 1.30±0.36 |
| BS14 | 0.13±0.01 | 0.84±0.17 | 3.79±0.99 | 0.28±0.25 | 3.81±1.59 | 2.06±1.06 |
| **BS15** | **0.34±0.01** | **2.85±0.03** | **8.37±0.44** | **0.95±0.05** | **9.88±0.56** | **8.00±0.36** |
| **BS19** | **0.10±0.01** | **1.67±0.15** | **8.79±0.97** | **1.30±0.18** | **8.53±1.35** | **6.04±0.50** |
| BS20 | 0.04±0.00 | 0.37±0.09 | 0.59±0.28 | 0.00±0.00 | 0.36±0.15 | 0.00±0.00 |
| BS21 | 0.14±0.09 | 2.75±0.89 | 5.35±2.76 | 0.76±0.61 | 7.06±4.80 | 3.83±2.44 |
| BS22 | 0.00±0.00 | 0.30±0.01 | 0.61±0.06 | 0.00±0.00 | 0.35±0.02 | 0.07±0.01 |
| BS23 | 0.03±0.03 | 0.70±0.08 | 0.86±0.06 | 0.00±0.00 | 1.05±0.02 | 0.00±0.00 |
| **BS24** | **0.18±0.02** | **1.75±0.26** | **10.13±1.55** | **0.97±0.24** | **8.35±1.07** | **4.15±0.31** |
| BS25 | 0.18±0.07 | 1.18±0.43 | 5.48±0.97 | 0.61±0.06 | 5.55±0.90 | 3.31±0.47 |
| BS32 | 0.21±0.01 | 1.28±0.05 | 4.87±0.04 | 0.32±0.01 | 3.28±0.12 | 2.08±0.07 |
| BS34 | 0.14±0.05 | 0.82±0.68 | 5.15±4.67 | 0.55±0.59 | 3.77±3.51 | 2.75±2.79 |
| BS35 | 0.22±0.01 | 1.16±0.07 | 4.05±0.14 | 0.26±0.01 | 3.15±0.08 | 1.72±0.03 |
| BS36 | 0.21±0.00 | 1.41±0.17 | 5.13±0.42 | 0.36±0.10 | 3.78±0.40 | 2.39±0.34 |
| BS37 | 0.22±0.01 | 1.02±0.05 | 3.37±0.20 | 0.18±0.03 | 2.24±0.10 | 1.15±0.09 |
| BS38 | 0.17±0.01 | 1.02±0.05 | 3.35±0.14 | 0.17±0.03 | 2.24±0.25 | 1.14±0.15 |
| BS39 | 0.07±0.02 | 0.29±0.12 | 0.34±0.31 | 0.00±0.00 | 0.89±0.83 | 0.57±0.53 |
| BS42 | 0.09±0.02 | 1.61±0.16 | 0.26±0.07 | 0.00±0.00 | 0.00±0.00 | 0.00±0.00 |
| **BS43** | **0.30±0.06** | **2.39±0.53** | **8.44±2.59** | **0.67±0.26** | **6.00±1.80** | **4.20±1.36** |
| BS45 | 0.06±0.00 | 0.69±0.33 | 0.07±0.07 | 0.00±0.00 | 0.00±0.00 | 0.00±0.00 |

BS23 did not synthesize D and F isoforms; BS22 did not synthetise A,D, BS39 -D, and BS42 as well as BS45—D,E,F, which was probably related to the increased production of the isoforms B (BS42, BS45), C (BS20, BS22) and E (BS23, BS39); (Table 3). It should be noted, however, that strains characterized by the highest efficiency of surfactin biosynthesis, BS15 and BS19, reducing ST by ca. 53, 49% respectively, synthesized all 6 isoforms (Fig 5). A significant high correlation (r = 0.51) between the decrease in ST [%] of the medium and the concentration of the B isoform [μg/mg] (Fig 6, Table 2) suggests that its biosynthesis may significantly affect the reduction of surface tension.

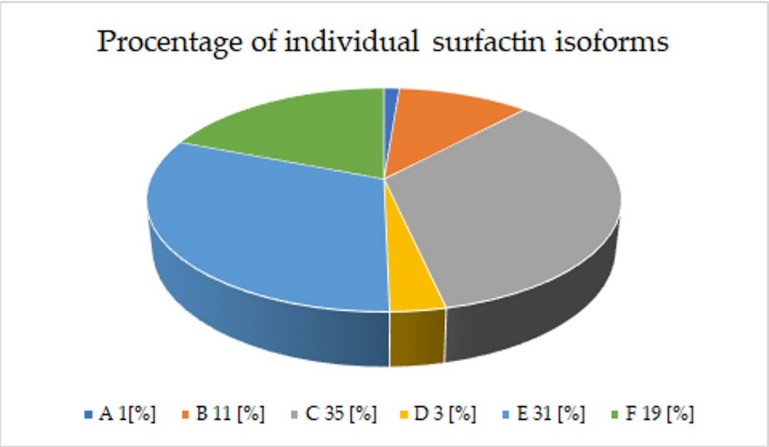

**Fig 5. Average percentage of individual isoforms of surfactin produced by selected *Bacillus subtilis natto* strains.**

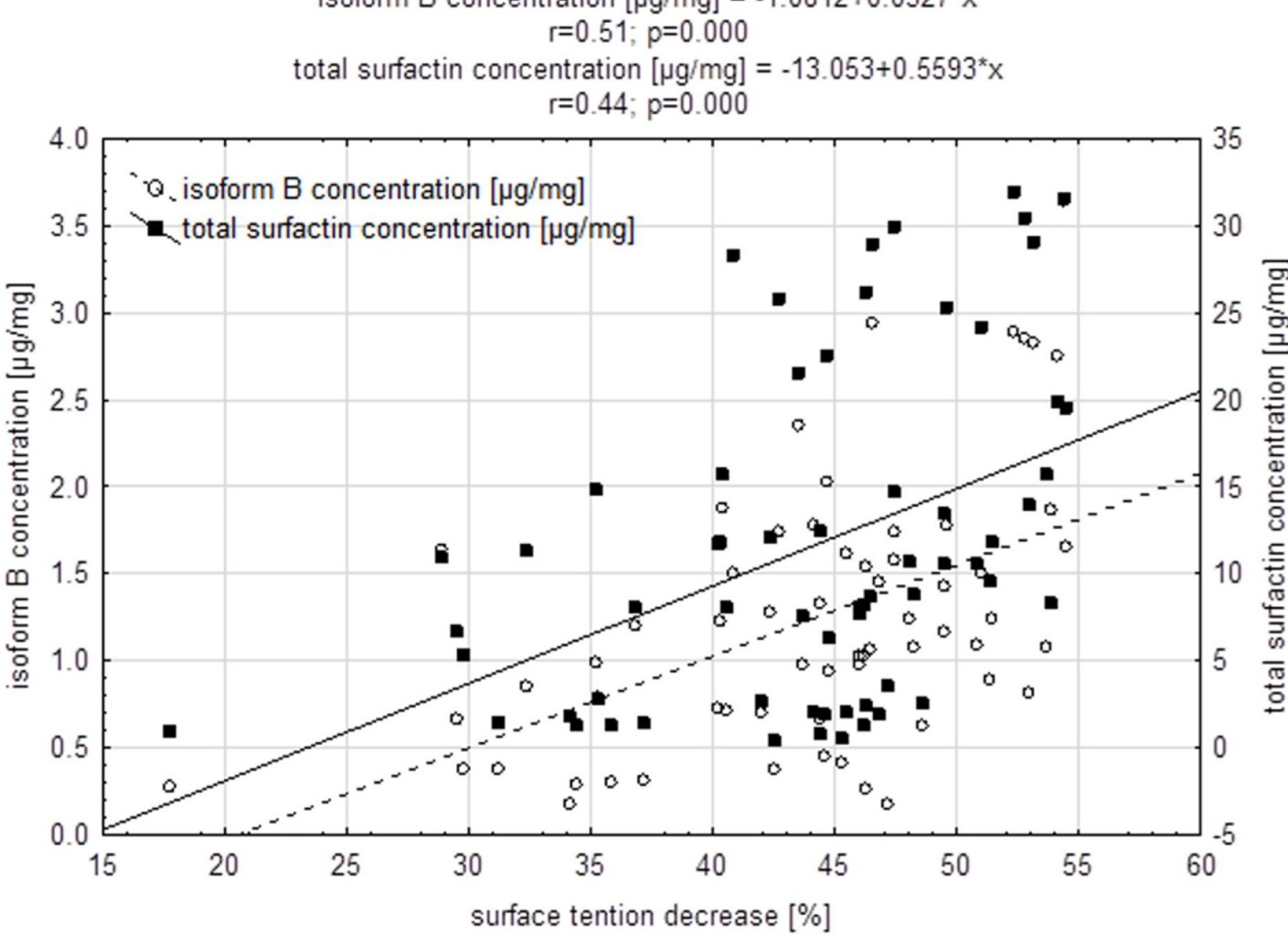

**Fig 6. Relationship between the decrease in surface tension [%] and the concentration of surfactin and its B isoform [µg/mg] in post-culture media of *Bacillus subtilis natto* strains (n = 63).**

This conclusion is also supported by the fact that all strains tested for surfactin production and its ability to reduce surface tension synthesized the B isoform, as opposed to the isoforms D, E or F. In addition, a very high percentage of this isoform, 82% and 84%, was found in the total concentration of surfactin produced by strains BS42 and BS45, respectively. Although these strains synthesized only three (A, B, C) of the six isoforms, they had the ability to lower the ST by as much as 45% (Table 1). Studies confirmed that the increased concentration of an individual isoform in total surfactin concentration can significantly determine the properties of the surfactant, including surface tension reduction [63]. The biosynthesis of an isoform containing 15 carbon atoms is particularly desirable, which has a CMC of 14.8 mg/L and can reduce water ST to 27.1 mN/m (25˚C) [64]. The surface and interfacial activity of surfactin increases with the length of the alkyl chain, thereby enhancing agregation of micelles. Liu et al. [65] also observed that an increase in the concentration of the isoform containing 15 carbon atoms in the alkyl chain increased oil-washing efficiency and oil displacement efficiency by surfactin synthesized by *Bacillus subtilis* BS-37. Razafindralambo et al. [66] reported a

relationship between the isoform consisting of 14 carbon atoms and the increased foam formation of the lipopetide, both in terms of foam density and liquid stability in foam. Thus, the structure of the peptide translates into its properties. Differences in the amount and concentration of synthesized isoforms may be genetically determined but also dependent on the composition of the culture medium and culture conditions (including pH, temperature, oxygenation) that affect the fermentation process [8,67]. For example, Jajor et al. [40] observed that reduced aeration of *Bacillus subtilis* KB1 and #309 cultures reduced the amount of analogues containing 15 carbon atoms and increased the amount of analogues with 12 atoms. This study can be extended to assess the impact of compound concentration in a two-phase (hydrophobic/hydrophilic) system on the shape of the surfactin peptide ring. It would also be possible to calculate the effect of biosurfactant addition on the interfacial tension in a two-phase system and to estimate lateral and rotational diffusion of the peptide ring [68]. Studies on the use of biomolecules such as surfactin can also be supplemented with modeling of molecular structure using density functional theory (DFT), which includes dispersion interactions and peptide bonds in peptide-based systems [69].

## Conclusions

The development of effective screening methods guarantees quick verification of isolated strains in terms of their ability to produce surfactin and thus creates the opportunity to identify efficient producers of this compound. Blood agar test can only be an initial evaluation criterion due to the low specificity of this method. In contrast, the measurement of surface tension during cultivation is a key parameter translating into the concentration of surfactin until the biosurfactant reaches its critical micellization concentration. Extraction using affinity chromatography with the SPE system significantly shortens the time of surfactin isolation from the culture medium and thus allows rapid verification of potential surfactin producers using the HPLC method. The three stages of screening presented in this study enable effective selection of surfactin producers. By this method we initially selected 45 strains capable of producing hemolysis zones, and then narrowed down the group to 19 strains significantly lowering the ST of the culture medium ($\geq$35%). The BS15 strain, which belonged to this group, reduced the surface tension of the culture medium by 52.77 $\pm$ 0.404 [%] and synthesized surfactin, including B isoform, at the highest concentration ($>$ 30 μg/mg; 2.85 $\pm$ 0.03 μg/mg, respectively). The selection of BS15 is additionally supported by the high repeatability of the results in the experimental replicates. For surfactin concentration, the standard deviation was 4.7%, and for the B isoform concentration standard deviation was only 1.05% of the mean. Thus, BS15 is a promising material for further research on the optimization of surfactin production.

## Author Contributions

**Conceptualization:** Beata Koim-Puchowska, Grzegorz Kłosowski, Dawid Mikulski.

**Formal analysis:** Beata Koim-Puchowska, Grzegorz Kłosowski, Dawid Mikulski, Aleksandra Menka.

**Funding acquisition:** Grzegorz Kłosowski.

**Investigation:** Beata Koim-Puchowska, Grzegorz Kłosowski, Dawid Mikulski.

**Methodology:** Beata Koim-Puchowska, Grzegorz Kłosowski, Dawid Mikulski.

**Project administration:** Grzegorz Kłosowski, Dawid Mikulski.

**Supervision:** Grzegorz Kłosowski.

**Validation:** Beata Koim-Puchowska, Dawid Mikulski.

**Writing – original draft:** Beata Koim-Puchowska.

**Writing – review & editing:** Grzegorz Kłosowski, Dawid Mikulski.

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
