## [Decision Letter · Decision Letter 0]

4 Oct 2019

PONE-D-19-24885

Evaluation of various methods of selection of B. subtilis strains capable of secreting surface-active compounds

PLOS ONE

Dear Grzegorz Klosowski, 

Thank you for submitting your manuscript to PLOS ONE. After careful consideration, we feel that it has merit but does not fully meet PLOS ONE’s publication criteria as it currently stands. Therefore, we invite you to submit a revised version of the manuscript that addresses the points raised during the review process.

We would appreciate receiving your revised manuscript by Nov 18 2019 11:59PM. To enhance the reproducibility of your results, we recommend that if applicable you deposit your laboratory protocols in protocols.io, where a protocol can be assigned its own identifier (DOI) such that it can be cited independently in the future. For instructions see: http://journals.plos.org/plosone/s/submission-guidelines#loc-laboratory-protocols

We look forward to receiving your revised manuscript.

Kind regards,

Ajaya Bhattarai

Academic Editor

PLOS ONE

**Journal Requirements:**

**Additional Editor Comments (if provided):**

In page-2, lines39-40, after CMC give reference. 

**Comments to the Author**

1. Is the manuscript technically sound, and do the data support the conclusions?

Reviewer #1: Yes

Reviewer #2: Yes

2. Has the statistical analysis been performed appropriately and rigorously? 

Reviewer #1: Yes

Reviewer #2: Yes

3. Have the authors made all data underlying the findings in their manuscript fully available?

Reviewer #1: Yes

Reviewer #2: Yes

4. Is the manuscript presented in an intelligible fashion and written in standard English?

Reviewer #1: Yes

Reviewer #2: Yes

5. Review Comments to the Author

Reviewer #1: In this submission to PLOS One, the authors carry out an experimental study on a three-step method for the selection of bacterial strains capable of producing surfactin. The authors find that their proposed selection strategy proved effective and made it possible to select the desired BS15 strain. The authors mention that the surfactin properties are derived from its chemical structure and is composed of a peptide loop of seven amino acids.

I consider this paper to be of interest to the PLOS One communities, and I am receptive towards publication subject to a minor comment. There has actually been prior work on computing peptides to understand surfactins, for example:

Biophysical Journal 85, 1377-1391 (2003)

Physical Chemistry Chemical Physics, 20, 18158-18168 (2018)

In principle, computation could also be used to complement the experiments investigated in this manuscript. I am not asking the authors to carry out any calculations; however, it should probably be mentioned that such studies could complement the experiments discussed in this work.

Reviewer #2: This manuscript assess different techniques for determining surfactin production in Bacilli. This paper is clearly written, and the work is comprehensive. I recommend publication. I wish to raise only a few minor points:

1. How did you determine that strains were different? Also, can you provide any additional information about these strains (e.g. 16S)?

2. No information is provided regarding submerged culture. Please provide more details.

3. The correlation between surfactin production and surface tension is not too strong. I wonder if a multivariate regression involving the different isoforms would yield a better correlation?

6. PLOS authors have the option to publish the peer review history of their article (what does this mean?). If published, this will include your full peer review and any attached files.

Reviewer #1: No

Reviewer #2: No

---

## [Author Response · Author response to Decision Letter 0]

25 Oct 2019

Detailed response to the reviewers' comments (PONE-D-19-24885)

We would like to thank the Editor and the Reviewers for their comments and suggestions. We have revised the manuscript point by point according to the Reviewer’s comments. All suggested changes are marked in yellow in the revised text and described below. 

We hope that the quality and readability of our manuscript has been improved.

Additional Editor Comments (if provided):

In page-2, lines 39-40, after CMC give reference.

The additional information has been added in line 40.

Response to Reviewer No. 1 comments:

I consider this paper to be of interest to the PLOS One communities, and I am receptive towards publication subject to a minor comment. There has actually been prior work on computing peptides to understand surfactins, for example:

Biophysical Journal 85, 1377-1391 (2003)

Physical Chemistry Chemical Physics, 20, 18158-18168 (2018)

In principle, computation could also be used to complement the experiments investigated in this manuscript. I am not asking the authors to carry out any calculations; however, it should probably be mentioned that such studies could complement the experiments discussed in this work.

The additional information has been added in lines 338-345, 566-570 according to the Reviewer suggestion.

Response to Reviewer No. 2 comments:

1. How did you determine that strains were different? Also, can you provide any additional information about these strains (e.g. 16S)?

The additional information about B. subtilis strain identification has been added in Isolation of Bacillus subtilis natto strains section (lines 128-130) according to the Reviewer suggestion. 

The isolated B. subtilis strains exhibited different metabolic features (a different ability to perform hemolysis, reduce surface tension and produce surfactin), which was also confirmed by MLST (multilocus sequence typing) analysis. Strains capable of producing the highest amount of surfactin (BS15, BS19, BS24, BS43) showed allele sequence variability in the pta (phosphate acetyltransferase) locus.

2. No information is provided regarding submerged culture. Please provide more details.

All SmF parameters i.e. composition of the medium, culture time, pH, inoculum size have been added in Submerged fermentation (SmF) conditions section (lines 131-141).

3. The correlation between surfactin production and surface tension is not too strong. I wonder if a multivariate regression involving the different isoforms would yield a better correlation?

The main aim of our work was to develop a method that would allow an effective and possibly fast selection of strains capable of surfactin biosynthesis. The use of the degree of surface tension reduction of the culture medium as one of the selection criteria for potential surfactin producers has significantly reduced the screening time. To this end, we determined the concentration of surfactin produced by strains that reduced surface tension by 33-35%. The value of the correlation coefficient between the percentage reduction in surface tension during the culture and the concentration of surfactin produced, which was about 0.5, indicated the existence of a relationship between these two variables. This confirms the correct choice of method for selecting surfactin producing strains. On this basis, we also conclude that after examining the other 24 strains for surfactin biosynthesis, the correlation coefficient value would increase. On the other hand, multiple regression does not seem to be the appropriate statistical tool for this study. For the estimators calculated by the method of least squares to exist and have the desired properties (estimators should be unbiased and efficient), certain assumptions must be met. One of them is the lack of collinearity of independent variables (concentrations of surfactin isoforms). In this study, the value of the correlation coefficient between independent variables is much higher than that between independent variables and the dependent variable, i.e. the percentage of surface tension decrease. In the case of D, E, F, C isoforms it is even ≥ 0.9. The use of a multiple regression model, explaining the variability of surface tension reduction as a result of biosynthesis of individual surfactin isoforms, would require enlarging the data set (which was beyond the scope of the work) and perhaps removing the most strongly collinear variables, i.e. catalysts.

---

## [Editor Report · Decision Letter 1]

30 Oct 2019

Evaluation of various methods of selection of B. subtilis strains capable of secreting surface-active compounds

PONE-D-19-24885R1

Dear Dr. Grzegorz Kłosowski,

We are pleased to inform you that your manuscript has been judged scientifically suitable for publication and will be formally accepted for publication once it complies with all outstanding technical requirements.

With kind regards,

Ajaya Bhattarai

Academic Editor

PLOS ONE
---

## [Editor Report · Acceptance letter]

5 Nov 2019

PONE-D-19-24885R1 

Evaluation of various methods of selection of *B. subtilis* strains capable of secreting surface-active compounds 

Dear Dr. Kłosowski:

I am pleased to inform you that your manuscript has been deemed suitable for publication in PLOS ONE. Congratulations! Your manuscript is now with our production department. 

With kind regards,

on behalf of

Dr. Ajaya Bhattarai 

Academic Editor

PLOS ONE